# Short-Term Metformin Therapy in Clomiphene Citrate Resistant PCOS Patients Improves Fertility Outcome by Regulating Follicular Fluid Redox Balance: A Case-Controlled Study

**DOI:** 10.3390/diagnostics14192215

**Published:** 2024-10-04

**Authors:** Mustafa Tas

**Affiliations:** Kayseri Acıbadem Hospital IVF-Unit, Kayseri 38030, Turkey; drmustafatas@yahoo.com

**Keywords:** clomiphene citrate, PCOS, redox balance, inflammation, follicle

## Abstract

Objectives: To determine the effect of short-term metformin administration on follicular fluid (FF) total oxidant status (TOS), total antioxidant status (TAS), oxidative stress index (OSI) and nuclear factor kappa B (NF-kB) in women with clomiphene citrate-resistant polycystic ovary syndrome (PCOS). Methods: Fifty-eight patients aged 23–34 who were planned to have intracytoplasmic sperm injection due to clomiphene citrate-resistant PCOS were included in the study. Participants were divided into two groups according to whether they used metformin or not. While 30 of 58 PCOS patients were using short-term metformin in combination with controlled ovarian stimulation, 28 PCOS patients were not using metformin. Metformin was started in the mid-luteal period and continued until the day before oocyte retrieval at 850 mg twice daily. To determine FF-NF-kB, TAS, TOS and OSI values, a dominant follicle ≥17–18 mm in diameter was selected for aspiration. Results: The number of mature follicles and fertilization rates of the metformin group were significantly higher than those not taking metformin. FF-TOS and OSI of the metformin group were significantly lower than those of the group not receiving metformin. Patients receiving metformin had higher FF-TAS than the group not receiving metformin. FF-NF-kB levels of the metformin group were significantly lower than the group not receiving metformin. Insulin resistance, FF-NF-kB and FF-TOS were negatively correlated with the number of mature oocytes. FF-TAS was positively correlated with the number of oocytes. Conclusions: Short-term metformin treatment in clomiphene-resistant PCOS improves the number of mature follicles and fertilization rates by regulating the intra-follicle redox balance.

## 1. Introduction

Polycystic ovary syndrome (PCOS), which becomes evident with ovulatory dysfunction, hyperandrogenism, irregular menstrual cycles and infertility, is seen in approximately 5–10% of women of reproductive age [1]. Local and systemic androgen increase induced by insulin resistance and compensatory hyperinsulinemia determine the phenotypic features of the disease and the changes in oocyte developmental capacity [2]. Despite the high number of oocytes collected, cycle cancelation and decreased fertilization rates are among the problems awaiting solution in PCOS [3,4].

Clomiphene citrate (CC) is the most preferred first-line drug to stimulate ovulation in infertile anovulatory PCOS patients. However, the moderate increase in pregnancy rates and the development of resistance to clomiphene in some patients necessitate the use of assisted reproductive treatments [5]. The GnRH antagonist protocol has come to the fore due to its lower risk of ovarian hyperstimulation syndrome (OHSS) and improvement in pregnancy rates [6]. However, despite the GnRH antagonist protocol, the moderate increase in clinical pregnancy rates has brought about the addition of new treatment agents to the protocol. The relationship between insulin resistance, hyperandrogenemia and anovulation in PCOS has enabled the long and/or short-term use of insulin sensitizing agents in stimulation protocols [7].

Metformin is an oral insulin sensitizer that, when added to the GnRH antagonist protocol, increases the sensitivity of the ovaries to gonadotropins and reduces the risk of OHSS [8]. Metformin is a biguanide that has an antihyperglycemic effect without causing hypoglycemia and increases insulin sensitivity in peripheral tissues and the ovaries [1,3]. It has been reported that metformin administration not only reduces androgen excess and insulin resistance in clomiphene-resistant PCOS, but also improves the ovarian response to gonadotropins [9].

Metformin acts mainly by complex I inhibition and protein kinase activation. Complex I functions with NADH in the maintenance of the mitochondrial proton gradient. The starting point of this study was the report that metformin, in addition to its conventional mechanisms of action, provides redox balance between the cytosol and mitochondria of developing follicles [10,11]. This study was designed to analyze the effect of short-term metformin therapy on follicular fluid redox balance and pro-inflammatory cytokine production in women with clomiphene citrate-resistant PCOS undergoing a GnRH antagonist-assisted conception treatment cycle. We evaluated the effect of adding metformin to the GnRH antagonist protocol on follicular fluid redox markers by measuring total oxidant status (TOS), total antioxidant status (TAS), and oxidative stress index (OSI). We also analyzed the effect of metformin on pro-inflammatory cytokine production by measuring the follicular fluid level of NF-kB, a cellular regulator of inflammation [12].

## 2. Materials and Methods

Fifty-eight patients aged between 23 and 35 years, who were scheduled for IVF/ICSI due to clomiphene citrate-resistant PCOS at Private Kayseri Acıbadem Hospital IVF Unit between January 2024 and June 2024, were included in this case–controlled study. Despite the use of 200 mg/day clomiphene citrate, lack of follicular development on the 14th day ultrasound examination and blood progesterone level <2.5 ng/dL on the 21st day were considered clomiphene citrate resistance. Women who met at least two of the criteria for hyperandrogenemia, ovulatory dysfunction and polycystic ovarian morphology determined by the ESHRE/ASRM were diagnosed with PCOS [13].

PCOS participants who received assisted conception treatment with the GnRH antagonist protocol were divided into two groups according to whether they used metformin or not. While 30 of 58 PCOS patients were using short-term metformin in combination with controlled ovarian stimulation (FSH plus metformin), 28 PCOS patients were not using metformin (rFSH without metformin). Metformin treatment was started in the mid-luteal period in patients with regular menstrual cycles, and on the first day of progesterone withdrawal bleeding in patients with irregular menstrual cycles. Short-term metformin use was continued until the day before oocyte retrieval at 850 mg twice daily. The groups given and not given metformin were matched in terms of age, duration of infertility and BMI. Since some patients showed insulin resistance and some did not, the groups could not be matched in terms of insulin resistance. The decision on whether patients should receive metformin treatment, and the duration of treatment was made by consensus of the physician and the patient, independent of insulin resistance.

Age, BMI, antral follicular count (AFC) and endometrial thickness on the day ovulation was triggered, as well as other reproductive parameters, were recorded in both groups. Basal hormone values and biochemical parameters were measured in blood samples taken after an 8–10 h overnight fast on the third day of spontaneous or progesterone withdrawal bleeding. Serum luteinizing hormone (LH), follicular stimulating hormone (FSH), insulin and total testosterone levels were determined by chemiluminescence method with cobas e601 immunoassay autoanalyzer (Roche Diagnostics GmbH, Mannheim, Germany). Fasting blood glucose measurement was performed by the hexokinase method with AU680 autoanalyzer (Beckman Coulter, Inc., Brea, CA, USA). Insulin resistance was calculated by homeostasis model assessment (HOMA-IR = insulin (μU/mL) × glucose (mg/dL)/405) [14].

GnRH antagonist protocol was administered to patients in both groups. Recombinant FSH (rFSH) treatment was started on the 2nd or 3rd day of spontaneous or progesterone withdrawal bleeding (Gonal-F, Merck Pharmaceutical Group Inc., Ataşehir, Turkey). rFSH dose adjustment was made taking into account patient age, ovarian reserve, BMI and ovarian response. When the dominant follicle was ≥14 mm, GnRH antagonist was started for pituitary suppression (Cetrotide 250 mg, Merck Serono, Istanbul, Turkey). Ovulation was triggered with a GnRH agonist triptoreline acetate (Gonapeptyl 0.1 mg/mL, Ferring, Istanbul, Turkey) when two or more follicles were ≥18 mm. Oocytes were collected 35–36 h after ovulation triggering. All metaphase II oocytes were subject to ICSI and eligible embryos were vitrified. Fertilization rate was determined as the number of fertilized oocytes per number of microinjected oocytes.

Women with congenital adrenal hyperplasia, type 2 diabetese mellitus, androgen-secreting tumors, Cushing’s syndrome, high thyroid stimulating hormone or prolactin levels were excluded from the study. Those who used anti-inflammatory, antioxidant and lipid-lowering drugs that would affect redox balance and cytokine production in the last three months were not included in the study. Those with a history of systemic inflammatory disease, endometrioma or hydrosalpinx were also excluded from the study. Those who took insulin sensitizers such as metformin, thiazolidinedione, inositol and berberine in the last three months were not included in the study due to the possibility of affecting the results.

### 2.1. Follicular Fluid Sampling

To determine each patient’s follicular fluid NF-κB, TAS, TOS and OSI values, a dominant follicle ≥17–18 mm in diameter and containing MII was selected for aspiration. Fluid belonging to the first follicle in the right ovary of each patient (selected at random) that met the dominant follicle requirement and was easy to reach was collected. Follicular fluids from other follicles in the right or left ovary were not evaluated. Since the molecular dynamics of each follicle and the nuclear maturation of the oocyte will be different, follicular fluids were not pooled. Intra-follicle flushing was avoided due to the risk of possible changes in redox balance markers. A different Eppendorf tube was used for each follicular fluid collected. If there were MII in the follicle aspirate, FF was evaluated, while fluids containing MI or GV oocytes were not evaluated. After the cumulus-granulosa cells in the FF were removed with hyaluronidase enzyme, they were centrifuged at 3000× *g* for 3 min and the supernatant was stored at −20 °C.

### 2.2. Follicular Fluid TAS, TOS, OSI and NF-kB Measurement

#### 2.2.1. FF-TAS and FF-TOS Analysis

Follicular fluid TAS and TOS levels were measured on an AU680 analyzer (Beckman Coulter, Inc., Brea, CA, USA) with the help of commercial kits (Rel Assay, Mega Medicine Industry & Trade Co., Gaziantep, Turkey). OSI was calculated by dividing the TOS value to the TAS values as a percentage [(TOS/TAS × 10). TOS was given as μmol H_2_O_2_ equiv/L, and TAS was given as mmol Trolox equiv/L. OSI results were given as an arbitrary unit [15,16].

#### 2.2.2. ELISA Analysis of FF-NF-kB

Follicular fluid NF-kB levels were measured by the quantitative sandwich enzyme immunoassay principle in accordance with the kit procedures (Sunred Biotechnology, Shanghai, China). The measurement range of the NF-kB kit was 0.15 ng/mL–40 ng/mL, and the minimum measurable level was 0.146 ng/mL. The intra- and inter-assay CV values of the NF-kB human ELISA kit were <10% and <12%, respectively. Plate washing was performed on a Biochrom Anthos Fluido 2 (Biochrom Ltd., Cambridge, UK). After the absorbances were measured at a wavelength of 450 nm on the CLARIOstar PLUS (BMG Labtech, Ortenberg, Germany) device, the concentrations corresponding to the absorbances were calculated.

### 2.3. Statistical Analysis

IBM SPSS Statistics Version 27.0 for Windows (IBM Corp., Armonk, NY, USA) was used for data analysis, and GraphPad Prism 8.0 (GraphPad Software Inc., San Diego, CA, USA) was used for graph drawing. Statistical significance between two independent groups that were not normally distributed was analyzed with the Mann–Whitney U test. Statistical significance between two normally distributed independent groups was analyzed by independent sample *t*-test. Categorical variables were analyzed with the chi-square test. The relationship between variables was made with Spearman correlation tests. Data are presented as mean ± standard deviation and median (1st quartile–3rd quartile) for continuous variables, and frequency (percentage) for categorical variables. The “Metan” R package was utilized for correlation matrix [17]. Two-tailed *p* values of <0.05 were considered significant.

## 3. Results

As shown in Table 1, age, BMI, infertility duration, AFC, the number of total follicles collected, serum testosterone, FSH, LH and HOMA-IR values were similar in patients taking metformin and those not taking it. The number of mature follicles in the metformin group (17 (16–18) vs. 13 (12–14), *p* < 0.001) was significantly higher than those not taking metformin. Similarly, the fertilization rates of the metformin group were significantly higher than those not taking metformin (76.5% vs. 38.5%, *p* = 0.035).

As shown in Table 2 and Figure 1, follicular fluid TOS level of the metformin group was significantly lower than the group not receiving metformin (3.65 ± 0.46 ng/mL vs. 5.79 ± 0.69 ng/mL, *p* < 0.001). Conversely, the FF-TAS level of patients taking metformin was higher than the group not taking metformin (1.02 ± 0.16 mmol Trolox Eq/L. vs. 0.28 ± 0.05 mmol Trolox Eq/L., *p* < 0.001). FF-OSI value in the metformin group was significantly lower than in the group not receiving metformin (0.36 ± 0.06 vs. 2.14 ± 0.44, *p* < 0.001). FF-NF-kB levels of the metformin group were significantly lower than the group not receiving metformin (5.83 ± 1.58 ng/mL vs. 12.86 ± 1.91 ng/mL, *p* < 0.001).

As seen in the correlation matrix in Figure 2, a negative correlation was detected between FF-TOS and the number of mature oocytes (r = −0.783, *p* < 0.001). A positive correlation was detected between FF-TAS and the number of mature oocytes (r = 0.341, *p* < 0.01). A negative correlation was detected between FF-OSI and the number of mature oocytes (r = −0.549, *p* < 0.001). A negative correlation was detected between FF-NF-kB and the number of mature oocytes (r = −0.485, *p* < 0.001). HOMA-IR negatively correlated with mature oocytes (r = −0.554, *p* < 0.001). FF-TOS and FF-NF-kB showed a positive significant correlation (r = 0.833, *p* < 0.001). FF-NF-kB showed a negative correlation with TAS (r = −0.734, *p* < 0.001) and a positive correlation with OSI (r = 0.799, *p* < 0.001) and HOMAIR (r = 0.322, *p* < 0.05). HOMA-IR was positively correlated with FF-TOS (r = 0.489, *p* < 0.001) and negatively correlated with total follicle collected (r = −0.297, *p* < 0.05). FF-TOS was negatively correlated with FF-TAS (r = −0.714, *p* < 0.001) and positively correlated with OSI (r = 0.857, *p* < 0.001). FF-TAS and FF-OSI were negatively correlated (r = −0.941, *p* < 0.001).

## 4. Discussion

Although heterogeneous results have been reported, such as not affecting, decreasing or increasing clinical pregnancy and live birth rates, in our clinical practice we often witness that PCOS patients have difficulty in becoming pregnant and maintaining pregnancy [18,19]. The increase in the total number of oocytes associated with high ovarian reserve may trigger the development of compromised oocytes, causing cycle cancelation or a decrease in fertilization rates [3,4]. Since PCOS is a multifaceted endocrine pathology, phenotypic variables are likely to negatively affect oocyte developmental capacity. Short- or long-term metformin treatment is the most commonly used protocol to improve oocyte developmental capacity in infertile PCOS patients [1,20]. In this context, adding medications to improve the metabolic state to conventional induction protocols in clomiphene-resistant PCOS patients has led to an improvement in fertility outcomes [21].

Although adding metformin to controlled ovarian stimulation is thought to improve reproductive parameters in women with CC-resistant PCOS by regulating intrafollicular dynamics, improving insulin resistance and hyperandrogenemia, there is no clearly established mechanism [20,21,22]. The demonstration that metformin acts by regulating cellular redox balance in many peripheral tissues suggested that it may improve fertility outcome in PCOS patients with the same mechanism of action [10,22]. This is the first clinical study to compare the effect of adding short-term metformin to GnRH antagonist protocol on follicular fluid redox balance markers TAS, TOS and OSI, as well as total number of oocytes, mature oocytes and fertilization rates in CC-resistant PCOS patients. While the antioxidant status indicator FF-TAS was four times higher in the metformin group than in the control group, the oxidant stress indicator TOS was two times lower. This finding is a critical indicator that metformin changes the follicular fluid redox balance in an antioxidant direction with its ROS scavenging effect. Moreover, the fact that the cumulative oxidative stress indicator OSI is 6 times higher in patients not taking metformin than in those taking metformin is another indicator that short-term metformin positively affects oocyte developmental capacity through a redox-dependent mechanism.

Although the role of oxidative stress (OS) in the etiology of PCOS is not clearly known, it has been reported that obesity, IR, compensatory hyperinsulinemia, hypernadrogenemia and hyperglycemia are strong stimuli of oxidative stress. However, the fact that ROS levels are high in women with PCOS who are non-hyperandrogenetic and do not have IR is evidence that factors other than phenotypic parameters contribute to the triggering of low-grade chronic inflammation and OS in PCOS [23]. Although we did not measure follicular fluid androgen and insulin levels, metformin reduces androgens through thecal cells and improves IR [21,24]. While HOMA-IR and FF-TAS show a negative correlation, the positive correlation between FF-TOS and IR supports the importance of IR in disrupting the follicular fluid redox balance. As a result, in addition to improving IR and hyperandrogenemia, advancing the redox balance in the antioxidant direction may be the possible mechanism by which metformin improves reproductive parameters in CC-resistant PCOS patients.

Follicular fluid TAS and TOS evaluation are effective and reliable markers in determining the redox balance of the follicle [25]. Since FF-TAS measurement shows both endogenous and exogenous antioxidant capacity, it provides more accurate results than measuring antioxidants one by one [26]. In addition to being a drug with antioxidant properties, metformin regulates the redox balance between the cytosol and mitochondria [10], which may be responsible for the increase in FF-TAS and the decrease in FF-TOS and OSI. One of the possible reasons for the higher number of mature and fertilized oocytes in the metformin group may be the increase in TAS levels. Metformin may provide this by increasing the cytosolic redox state [NADH]: [NAD+] [27]. The fact that metformin and some insulin sensitizing agents contribute to oocyte developmental capacity by reducing androgen-induced endoplasmic reticulum stress in granulosa cells, as well as autophagy regulation, supports that this drug acts through the redox balance in the follicular microenvironment [25,28,29]. Another possible explanation may be that metformin contributes to the acquisition of mature and fertilized oocytes by reducing oxidative phosphorylation and pro-inflammatory cytokine and ROS production [12,26]. The positive correlation of FF-NF-kB, a marker of cellular inflammation, and TOS levels is another finding that supports that metformin causes improvement in fertility outcome by blocking the NF-κB pathway.

An important limitation of the study is that it did not analyze clinical pregnancy and live birth rates and did not categorize PCOS patients according to phenotypes. By separating patients according to phenotypes, it may be possible to elucidate how syndrome-specific endocrine and metabolic pathologies affect FF redox balance and fertility outcome. Adding short-term metformin treatment to improve follicle quality after the first ICSI attempt in CC-resistant PCOS patients may positively affect fertility outcome. A clearer discussion can be reached with studies using long-term metformin.

## 5. Conclusions

The current study is important in that it provides the first clinical data investigating the effects of adding short-term metformin to GnRH antagonists on follicular fluid redox markers and pro-inflammatory cytokine production in CC-resistant PCOS patients. The higher number of mature oocytes and fertilized oocytes in the group administered metformin supports that this drug positively affects oocyte developmental capacity. Despite the positive correlation between FF-TAS and the number of mature and fertilized oocytes, the negative correlation between FF-TOS and OSI and the number of mature and fertilized oocytes suggests that metformin improves folliculogenesis through a redox-dependent mechanism of action. The fact that IR reduces FF-TAS levels while causing an increase in FF-TOS and OSI is evidence that one of the effects of metformin in increasing oocyte developmental capacity is the decrease in IR. In order to increase the number of mature and fertilized oocytes in CC-resistant PCOS patients, short-term metformin administration in appropriate cases may improve fertility outcome by regulating the redox balance.

## Figures and Tables

**Figure 1 diagnostics-14-02215-f001:**
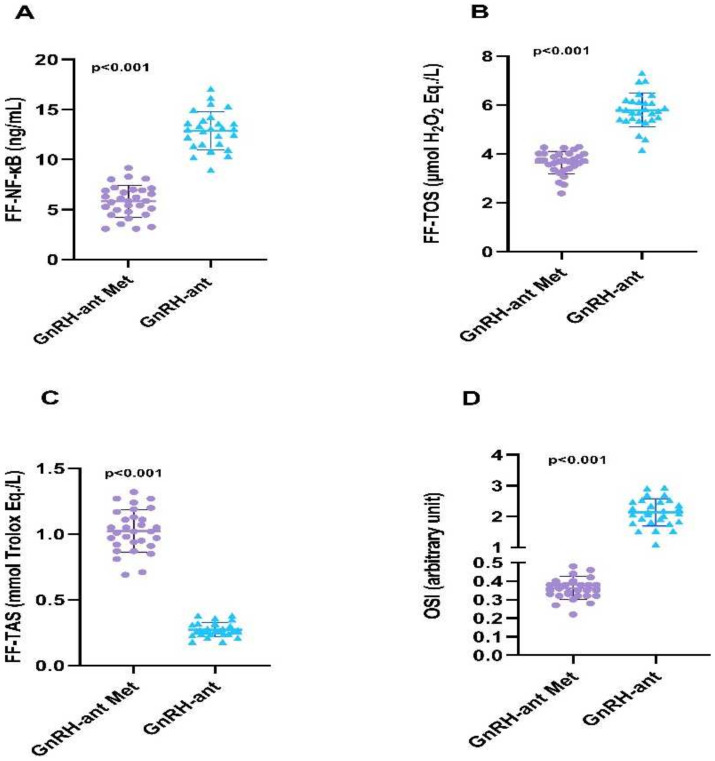
Graphical representation of follicular fluid NF-kB (**A**), TOS (**B**), TAS (**C**) and OSI (**D**) levels in groups that received and did not receive short-term metformin treatment. Note that inflammatory markers (NF-KB) and oxidant markers (TOS, OSI) decreased in the metformin group, while antioxidant TAS increased.

**Figure 2 diagnostics-14-02215-f002:**
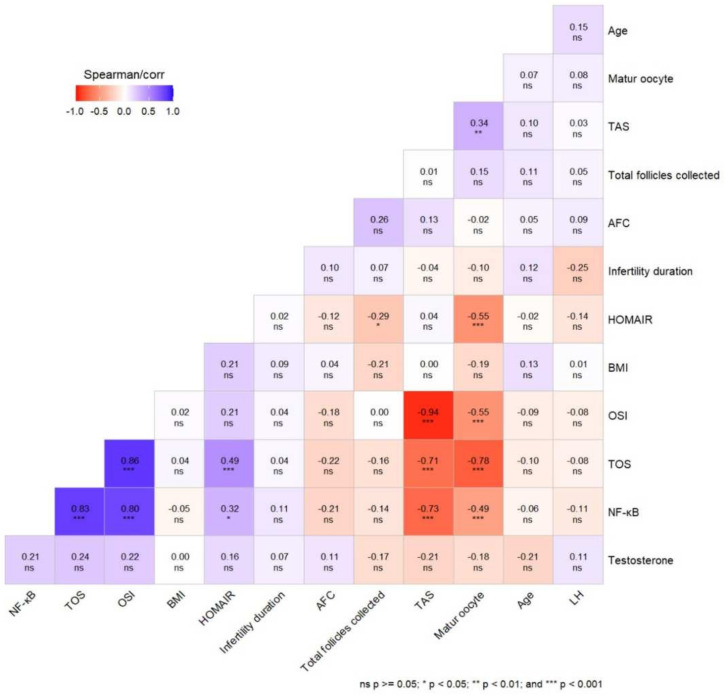
Graphical representation of correlation matrix of variables. Blue colors indicate positive correlations, and red colors indicate negative correlations.

**Table 1 diagnostics-14-02215-t001:** Comparison of demographic and hormonal characteristics of participants treated with GnRH antagonist protocol with and without metformin.

Variables	GnRH Antagonist Plus Metformin (*n* = 30)	GnRH Antagonist without Metformin (*n* = 28)	*p*-Values
Age (years)	29 (26–33)	30.5 (26–32)	0.574 **
BMI (kg/m^2^)	24.63 ± 1.79	24.47 ± 1.81	0.736 *
Infertility duration (yrs)	4 (3–5)	4 (3.25–5)	0.674 **
Antral follicle count	25 (23–27)	24 (23–25)	0.131 **
Total oocytes collected	22.00 ± 2.05	21.61 ± 1.57	0.419 *
Total ooctes collected	21.17 ± 2.21	20.82 ± 1.52	0.489 *
Matur oocytes (MII)	17 (16–18)	13 (12–14)	<0.001 **
Fertilization rate (*n*, %)	13 (76.5%)	5 (38.5%)	0.035 ***
Testosterone (ng/dL)	37.46 ± 10.7	40.9 ± 8.77	0.188 *
FSH (mIU/mL)	5.64 ± 1.15	5.92 ± 1.20	0.370 *
LH (mIU/mL)	10.8 (8.3–13.1)	9.7 (9.4–11.5)	0.613 **
HOMA-IR	2.08 ± 089	2.16 ± 063	0.706 *

* Student *t*-test, ** Mann–Whitney U test. *** Chi-Square test. BMI; body mass index, FSH; follicular stimulating hormone, LH; luteinizing hormone, HOMA-IR; Homeostasis Model Assessment of Insulin Resistance.

**Table 2 diagnostics-14-02215-t002:** Comparison of follicular fluid redox balance and pro-inflammatory cytokines concentrations in groups receiving and not receiving metformin.

Redox Balance Markers	GnRH Antagonist Plus Metformin (*n* = 30)	GnRH Antagonist without Metformin (*n* = 28)	*p*-Values ***
FF-TOS (μmol H_2_O_2_ equiv/L)	3.65 ± 0.46	5.79 ± 0.69	<0.001
FF-TAS (mmol Trolox equiv/L)	1.02 ± 0.16	0.28 ± 0.05	<0.001
FF-OSI (Arbitrary unit)	0.36 ± 0.06	2.14 ± 0.44	<0.001
FF-NF-kB (ng/mL)	5.83 ± 1.58	12.86 ± 1.91	<0.001

* Student *t*-test. FF; follicular fluid, TOS; total oxidant status, TAS; total antioxidant status, OSI; oxidative stress index.

## Data Availability

All data generated or analyzed during this study are included in this published article.

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
