# Peer review of "Short-Term Metformin Therapy in Clomiphene Citrate Resistant PCOS Patients Improves Fertility Outcome by Regulating Follicular Fluid Redox Balance: A Case-Controlled Study"

_diagnostics, 2024, doi:10.3390/diagnostics14192215_

Round 1
Reviewer 1 Report
Comments and Suggestions for Authors
Dear Authors
Thanks for your interesting study, There are some points which are better to consider,
1- Please mention the type of study design in the method section and manuscript title
2-Please describe the study performed in community clinics or academic hospitals.
Author Response
Reply for R1
Q1- Please mention the type of study design in the method section and manuscript title
R1- The type of study was highlighted in both the title and method sections.
Q2-Please describe the study performed in community clinics or academic hospitals.
R2: It was highlighted as a private hospital.
Reviewer 2 Report
Comments and Suggestions for Authors
This paper deepens into the mechanisms behind metformin treatment for CC-resistant PCOS patients. The article is generally well-written and organized, and the results present new insights behind the mechanism of action of metformin. Still, I have some questions/comments for the author to address:
1. I was surprised that only one author signed the manuscript, and there were no acknowledgments. For a manuscript that includes clinical data, laboratory work, and advanced statistics, I was sincerely surprised.
2. Comments about the Abstract:
- "Results" is repeated twice.
- What does HOMA-IR mean? The abstract gets a bit confusing with so many abbreviations. Also, this journal is not focused on gynecology, so I recommend defining PCOS and FSH so the reader can follow.
3. Comments about the Methods and Results:
- Why did the author include women only up to 35 years old?
- Was there a maximum/same amount of cycles with unresponsiveness to CC for all patients before considering them CC-resistant?
- What about patients treated with letrozole?
- Did the author measure redox balance markers' basal levels (before treatment)? And the hormonal characteristics? This comparison could be interesting to include.
- I recommend to complete Figure legends. They seem too brief for such complex figures, especially Figure nr 2.
4. Comments about the Discussion:
- The author postulates different reasons behind the positive effect of metformin. Do you have any suggestions of how this research line could be continued?
- What are the implications of these results?
Author Response
Reply for R2
Q1:I was surprised that only one author signed the manuscript, and there were no acknowledgments. For a manuscript that includes clinical data, laboratory work, and advanced statistics, I was sincerely surprised.
R1: We thank Dr. Meltem Yardım for her help in the analysis and interpretation of biochemical parameters.
Q2. Comments about the Abstract:
- "Results" is repeated twice. It was deleted
- What does HOMA-IR mean? The abstract gets a bit confusing with so many abbreviations. Also, this journal is not focused on gynecology, so I recommend defining PCOS and FSH so the reader can follow. All abbreviations are explained before their first use.
Q3. Comments about the Methods and Results:
- Why did the author include women only up to 35 years old? We determined the age of 35 as the upper limit because the redox balance deteriorates with age.
- Was there a maximum/same amount of cycles with unresponsiveness to CC for all patients before considering them CC-resistant? - To be considered as CC resistant, a history of at least two cycles of CC non-response was required.
- What about patients treated with letrozole? We did not include those receiving letrozole treatment in the study.
- Did the author measure redox balance markers' basal levels (before treatment)? And the hormonal characteristics? This comparison could be interesting to include. Since these patients are stimulated cycle patients, it is technically not possible to measure the basal levels (before treatment) of redox balance markers.
- I recommend to complete Figure legends. They seem too brief for such complex figures, especially Figure 2. They were expanded.
Q4. Comments about the Discussion:
- The author postulates different reasons behind the positive effect of metformin. Do you have any suggestions of how this research line could be continued? The sentence "A clearer discussion can be reached with studies using long-term metformin" was added to the discussion section.
- What are the implications of these results? Adding short-term metformin treatment to improve follicle quality after the first ICSI attempt in CC-resistant PCOS patients may positively affect fertility outcome.
Reviewer 3 Report
Comments and Suggestions for Authors
The research Focuses on the potential of metformin to improve fertility results in women with PCOS resistant to clomiphene citrate treatment. This study offers insights into both the implications and the biological processes that underlie metformins impact, on reproductive success. The research involved a number of 58 individuals split into two groupsâone that received metformin, with stimulation and another that did not receive metformin for comparison purposes.A systematic approach was used to assess the impact of metformin on fertility outcomes.The detailed examination of markers in fluid, like oxidant status total antioxidant status and oxidative stress index enhanced the studys conclusions.
Polycystic ovary syndrome often leads to infertility issues and resistance, to citrate can make treatment more challenging for women struggling with this condition. The discovery that metformin not boosts the count of follicles but also enhances fertilization rates is a crucial breakthrough for healthcare providers treating PCOS patients. Introducing metformin as a supplement to gonadtropin therapy. In the term. May represent a significant advancement in improving fertility outcomes, for women dealing with PCOS.
The researchs attention, to maintaining a balance of redox in fluid to enhance fertility outcomes is groundbreaking in its approach. The study showcases metformins ability to decrease stress and inflammation beyond regulating insulin sensitivity. Moreover the decrease in inflammatory cytokines such as NF kB underlines metformins promising therapeutic effects, on enhancing the quality of oocytes. One issue is the absence of long term data, in our analysis. The research primarily looks into the impacts of short term metformin treatment. Does not delve into the lasting reproductive or metabolic consequences of metformin, on individuals with PCOS ( ovary syndrome). It would be valuable, for upcoming studies to consider assessing not pregnancy rates but also live birth rates and the durability of the enhanced results. Polycystic ovary syndrome is a condition, with manifestations in individuals.The research does not categorize patients according to PCOS traits, like those lacking insulin sensitivity or high androgen levels.It would be valuable to investigate how various PCOS patient subgroups react to metformin treatment as its effectiveness may differ based on these characteristics.
Even though the research made sure that patients were similar, in terms of age and BMI as how long they had been dealing with infertility issues; it pointed out that insulin resistance was not evenly spread among the different groups.It is important to note this because insulin resistance is crucial, in understanding PCOSs underlying processes.Future studies should aim to manage this influencing factor carefully.
The article offers information, about using short term metformin therapy in PCOS patients who don't respond to clomiphene treatment It suggests that metformin can help enhance fertility outcomes by balancing levels in follicular fluid and decreasing oxidative stress. This study makes a significant contribution to the field of reproductive medicine and can be beneficial for doctors looking for new options, for patients who don't respond well to standard ovulation induction treatments despite a few drawbacks.
Author Response
Reply for r3
Thank you very much for your valuable comments. In line with your suggestions, a limitation paragraph has been added to the Discussion section. “An important limitation of the study is that it did not analyze pregnancy and live birth rates and did not categorize PCOS patients according to phenotypes. By separating patients according to phenotypes, it may be possible to elucidate how syndrome-specific endocrine and metabolic pathologies affect FF redox balance and fertility outcome”.
A new reference was added (29)